# Risk assessment of pesticide residues ingestion in food offered by institutional restaurant menus

**Thuany Matias da Silva[1], Larissa Mont'Alverne Jucá Seabra[1], Luciléia Granhen Tavares Colares[2], Barbara Lettyccya Pereira Chacon de Araújo[1], Vanessa Cristina da Costa Pires[1], Priscilla Moura Rolim[1]***

1 Department of Nutrition, Federal University of Rio Grande do Norte, Natal, Rio Grande do Norte, Brazil,
2 Department of Nutrition, Federal University of Rio de Janeiro, Rio de Janeiro, Rio de Janeiro, Brazil

* priscillanutri@hotmail.com

**Data Availability Statement:** All relevant data are within the manuscript and its Supporting information files.

## Abstract

The chronic ingestion of pesticide residues through food appears to be a global public health issue, especially in Brazil. This study evaluates 120 menus across six Brazilian institutional restaurants, estimating the allowance of active pesticide ingredients, residue characterization, and chronic exposure risk through food. Data analysis reveals 263 authorized active ingredients, predominantly insecticides (43%), fungicides (40%), and herbicides (14%) for use in 40 foods. Notably, 4% of residues are extremely toxic, 5% highly toxic, and 14% moderately toxic. Forty-two compounds, especially those permitted in animal-source foods, exhibit high bioaccumulation potential. Some foods harbor multiple pesticide residues, raising concerns, despite 99% of residues falling within the Acceptable Daily Intake. Methomyl insecticide poses potential ingestion risks during lunch, warranting attention. The pervasive presence of pesticide residues in daily consumed foods underscores the necessity for greater attention to the source of the food, ensuring access to healthy and safe collective consumption.

## 1. Introduction

The large-scale production of commodities in Brazil consequently increased the use of pesticides, further exposing the population to risks caused by these harmful substances [1]. These substances are products and agents of physical, chemical, or biological processes intended for use in agricultural products' production and storage sectors, whose objective is to change the composition of fauna or flora to prevent harmful living beings [2]. Insecticides, herbicides, fungicides, and rodenticides [3], classified as extremely toxic to health, highly toxic, moderately toxic, slightly toxic, and unlikely to cause any harm to health, can be used in food production [4].

The insertion of pesticides in Brazil has a historical nature, with strong economic support since the Green Revolution [5], which is considered a milestone for the consolidation of the hegemonic agrochemical model, making the country one of the world leaders in

**Funding:** This work was funded by the Ministry of Education Foundation - Coordination for the Improvement of Higher Education Personnel (CAPES), code 001 awarded to PMR.

**Competing interests:** The authors have declared that no competing interests exist.

the consumption of pesticides [6, 7]. Brazil has already sold around 2,300 pesticide formulations, many of which are banned in the European Union (EU) [8]. A survey by the Food and Agriculture Organization of the United Nations shows that Brazil uses more pesticides on its crops than China and the United States combined. In 2021, 3.31 kg of pesticides per person were used in Brazil, while in the United States, 1.36kg per capita, and in China, 0.17 kg per capita [9].

Pesticides have harmful effects on the health of rural workers, the general population, and the environment [10]. They can cause serious health problems, including chronic and acute toxicities. Dermal and respiratory tract diseases are the most common acute effects of pesticide exposure, which are the most frequent ways they enter the human body [11]. Regarding chronic intoxications, which are exposed throughout life, neurological diseases, various types of cancer, disorders of the reproductive system, such as infertility, spontaneous abortions, fetal malformations, endocrine disruptors, as well as hematological and in development [12–15]

The toxicological evaluation of these substances commonly involves oral, acute skin, and acute inhalation toxicity, with studies performed over a short period, around three months of animal testing (90 days) [16]. Therefore, the concern regarding chronic exposure to these substances through food consumption is underscored, primarily because of the significant quantity of residues utilized and their cumulative and synergistic effects [17]. Thus, based on the precautionary principle, the population must be informed and alerted about the potential risks' pesticides can cause to protect their health and ecosystems [18]. Since the 2000s, Hamilton (2000) [19] posed the following question: "how much pesticide residue is present in my food today?" and the answer: "There should not be more residues in the food than unavoidable when the pesticide is used for effective pest control, and the amount of consumed residue should not be harmful to health." However, the study focuses on estimating likely residues in food with the purpose of predicting short-term intake of residues, using data from the international perspective of Codex Alimentarius.

The Pesticide Residue Analysis Program in Food (named PARA in Brazil), coordinated by the National Health Surveillance Agency (ANVISA) investigated 122 active ingredients (a.i) in foods and detected irregularities in 50% of the samples, including bell peppers, strawberries, cucumbers, and lettuce (Pesticide Residue Analysis Program in Food [PARA]. In the most recent report, 25.6% of the samples analyzed were non-compliant for some a.i such as carbofuran, ethephon, and formetanate, present in foods such as orange, papaya, grapes, kale, passion fruit, and bell peppers [20]. Data from the Brazilian Institute for Consumer Protection found pesticides in samples of ultra-processed foods derived from meat and milk, ultra-processed vegetables, such as soy drinks, breakfast cereals, snacks, tubes, water biscuits and crackers [21].

The indicators used to evaluate pesticide residues in food are the Maximum Residue Limit (MRL), referring to the maximum amount of pesticide residue allowed in food, the Acceptable Daily Intake (ADI), the estimated amount of the a.i present in foods that it can be ingested daily throughout life, without posing a risk to the consumer's health; and the indicator that estimates the Theoretical Maximum Daily Intake (TMDI), which is the maximum estimated amount of pesticide residue in food ingested per capita daily [22].

Institutional Restaurants (IR) are food environments that ensure food and nutritional security [23]. In this sense, indicators that assess the risk of food contamination by pesticides are relevant for food service management, particularly in planning and evaluating menus, seeking to adopt more sustainable practices in meal production [24]. Promoting healthy and sustainable diets is closely related to food systems guaranteeing the human right to adequate food [25, 26]. The relationship between consumption, supply, and availability of sufficient,

nutritious, safe, sustainable, and poison-free food that meets dietary needs is the basis for a population's food and nutrition security [27]

The food service sector plays a significant role in the socioeconomic context, as it produces an average of 35.5 million meals per day in Brazil [28] (Brazilian Association of Collective Meal Companies), contributing to the maintenance or recovery of the health of the clientele served. Furthermore, the nutritionist, the food service manager, is tasked with planning, organizing, supervising, and evaluating meal production. This role includes providing food and nutritional guidance and education to the community and implementing sustainable practices in meal preparation. These actions need to be integrated into food systems in an expanded view of food quality in terms of sustainability dimensions (Federal Council of Nutritionists) [23]. Furthermore, institutional food services represent a fundamental space in food acquisition, as they purchase large quantities of food and can be transformative agents for more sustainable food purchases.

The food production in Brazil is highly dependent on pesticides and unsustainable food systems, from the field to the consumer's plate [29]. Thus, the scarcity of data regarding the risks of exposure to pesticides through food, the absence of effective communication about the consumption of food contaminated with various pesticide residues, and the health impacts of these substances, especially concerning the cumulative and synergistic effects of these substances, highlight the significance of this study. In this sense, this research aims to evaluate the foods offered on menus of restaurants located in public educational institutions in terms of active ingredients authorized for use in food crops, as well as the risk of exposure to pesticide residues through estimated intake in a daily menu.

## 2. Material and methods

### 2.1. Study design

This is a cross-sectional, quantitative, and descriptive study. The study design can be seen in Fig 1.

### 2.2. Research site and data source

This study analyzed menu data from six institutional restaurants in public educational institutions in Rio Grande do Norte state, Brazil. The restaurants were selected using non-probability sampling for convenience. Menu data were collected in a previous study based on Standardized Recipe Sheets (SRS) used in restaurants.

### 2.3. Data collection

**2.3.1. Analysis of pesticides approved for use in food.** This study used the per capita food values from 120 lunch menus from six institutional restaurants previously collected by Nogueira et al. [30]. The study included restaurants serving customers between the ages of 18 and 60.

The list of pesticides (active ingredients) per plant-source foods, Brazilian MRL (mg/Kg) and ADI (mg/Kg body weight) were collected from the database of the National Health Surveillance Agency (ANVISA) (2021) and for animal-source foods or values not found in the database data from ANVISA [31], the Codex Alimentarius database [32] and the European Union database [33] was consulted. Data collection for active ingredients of pesticides occurred from September 2022 to March 2023.

Food items that did not have an MRL and AI that did not present an ADI were excluded from the study. The mixed seasoning, Worcestershire sauce, salt, meat tenderizer, fish, and

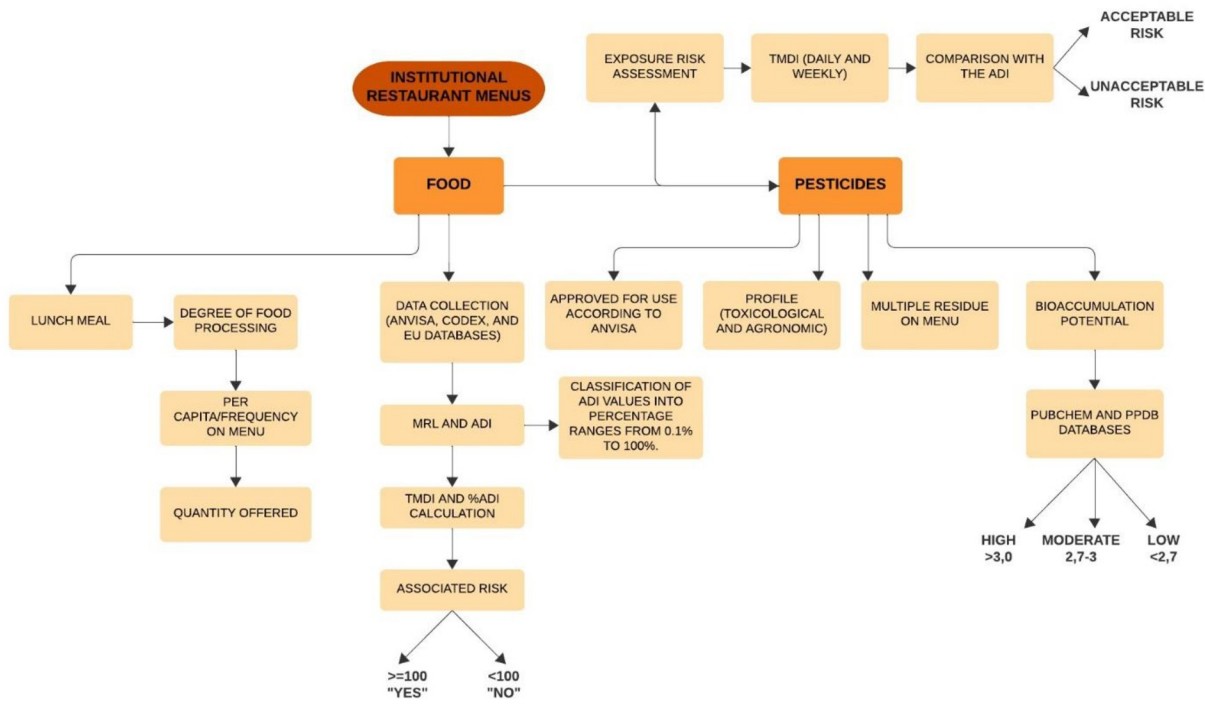

**Fig 1. Schematic representation of the research design.**

"cajá" (*Spondias mombin)* were also excluded from the study as they did not contain data. For certain types of processed and ultra-processed foods, as well as culinary ingredients like flour, corn starch, pasta, fruit pulp, butter, and cheese, the analysis was done based on the ingredients that represented the most significant quantity in the food. For example, the analysis for pasta was based on wheat, fruit pulp was based on fruit, orange was based on citrus fruit and cheese and dairy products were based on milk.

## 2.4. Active ingredients (AI) characterization

**2.4.1 Toxicological and agronomic profile.** The active ingredients used to assess the toxicological profile classification was the most frequently used in an average of the six restaurants. This classification considered the ANVISA [17] and the agronomic profile according to the Ministry of Agriculture and Livestock [34]. The contribution of each food group was assessed based on approved active ingredient amounts, and the presence of multiple pesticide residues in a single food was examined.

## 2.5. Bioaccumulation potential

To analyze the bioaccumulation potential, the PubChem [35] and Pesticide Properties Data-Base (PPDB) [36] were assessed. The AI's octanol-water partition coefficient values (Log P or Log Kow) in the foods on the menus were utilized for this purpose. The reference values used for evaluation were: Log P or Log Kow: < 2.7 = low bioaccumulation; 2.7–3 = Moderate bioaccumulation; > 3.0 = high bioaccumulation.

The n-octanol/water partition coefficient (Kow) defines the hydrophobic properties of molecules. Kow is the ratio value between the concentrations of a substance dissolved in a two-

phase system consisting of two immiscible liquids (n-octanol and water). The reason refers to the affinity in which water acts as the polar phase, whereas octanol acts as the nonpolar phase, reflecting the molecule's lipophilicity [37]

## 2.6. Acceptable daily intake (ADI) of the pesticide

The ADI values collected from the National Health Surveillance Agency and CODEX data-bases were classified into ranges to better analyze pesticide exposure based on %ADI. Ranges were established from 0.1 to >100% and distributed as follows: 0.1–10%; 10–20%; 20–30%; 40–50%; 50–60%; 70–80%; 80–90%; 90–100% and >100%.

## 2.7. Assessment of Theoretical Maximum Daily Intake (TMDI)

The calculations for estimating intake were based on the World Health Organization (WHO) reference guide document [22]. Initially, the food per capita values offered over the four weeks were summed up and divided by the frequency in which the food appeared on the menu, obtaining the average per capita value (in Kg). Subsequently, the average per capita value was multiplied by the respective MRL (mg/Kg) of the AI to obtain the TMDI value (mg/Kg), which indicates the person's degree of exposure to the given substance. The TMDI estimate was performed according to Eq 1.

**Equation 1**. TMDI Estimation:

$$TMDI = (MRL\ x\ average\ per\ capita) \tag{1}$$

MRL represents the maximum residue limit expressed in mg per kg of food and average food per capita obtained from SRS using information on the quantity and frequency in which they appeared on the menu, expressed in kg.

## 2.8. Risk of exposure from the perspective of the food

For risk analysis, the TMDI values were compared with the ADI to assess whether the AI presents an unacceptable or acceptable risk, as established by the WHO [22]. To this matter, the ADI percentage of each AI was calculated. The %ADI value made it possible to analyze the compounds in which the TMDI exceeded the ADI values. If the TMDI value was $>=100$, it was considered an unacceptable risk, according to the WHO. Eq 2 describes the %ADI calculation in which the TMDI value was multiplied by 100 and then divided by ADI based on body weight (kg).

**Equation 2**. % ADI:

$$\%\ ADI = \frac{TMDI\ x\ 100}{ADI\ x\ body\ weight} \tag{2}$$

The anthropometric data by Brazilian Family Budget Survey [38] was used as a reference for body weight, considering the age range of 18 to 60 years. The average weight used was 65 kg.

## 2.9. Estimation of pesticide intake in a daily lunch menu and exposure risk

For this analysis, 12 active ingredients (a.i) were selected and listed according to occurrence/ frequency of it on menus, degree of toxicity, and detection data in the Brazilian pesticide analysis program. For each IR, foods usually part of the daily lunch menu were exemplified (meat or chicken, rice, beans, salad and/or vegetables, fruit and/or juice). Then, the TMDI values for

each a.i. were summed up from Monday to Friday for the four weeks evaluated. An average TMDI was obtained.

According to WHO recommendations [22], the TMDI average and ADI values of each residue were compared to assess the risk they pose (acceptable or unacceptable), considering the body weight of 65kg. In cases where the TMDI values exceed the ADI, the risk level for the respective a.i. is considered unacceptable. To expand the analysis, making it as realistic as possible, an estimate of weekly intake (possible cumulative risk) was also carried out, considering the pesticide residue ingestion throughout the usual food consumption for a week. To this matter, the TMDI value of the respective a.i. was added together for four weeks and divided by four.

## 2.10. Data analysis

The database was created using Microsoft Excel® spreadsheets, version 2016. When necessary, descriptive statistical analyses, including relative frequency, mean values, and standard deviation, were performed.

## 2.11. Ethical considerations

Our research followed all ethical requirements established by the Resolution of the National Health Council No. 466/2012. All research protocols were submitted to the Ethics Committee of the Onofre Lopes University Hospital (CEP/HUOL) at the Federal University of Rio Grande do Norte (UFRN) and received approval under protocol number CAAE 92734418.5.0000.5292.

Additionally, letters of agreement were obtained with all signatures of the managers from all investigated evaluation units. All these documents are kept in the possession of the researcher. The on-site data collection regarding the menus offered in the food services of the evaluated institutions took place in the years 2018 and 2019. The calculations for estimating pesticide levels in the menus were conducted in 2023, and the databases utilized are from public domain official sources. Data on pesticide exclusion limits, acceptable daily consumption and anthropometric data of the population are in the public domain and available on official government websites and databases.

# 3. Results and discussion

## 3.1. Characterization of the foods offered on the menus

The food groups identified on the menus were cereals, tubers, legumes, vegetables, fruits, and animal-source food. It is worth highlighting that most foods on the menu were unprocessed and minimally processed. This result demonstrates that the menus were planned considering the recommendations of the Dietary Guidelines for the Brazilian population, which is an official document that addresses the principles and recommendations of an adequate and healthy diet for the Brazilian population [39]

From this perspective, the foods that are present on the menus, even though they are within the recommendations of the dietary guidelines and are in natura/unprocessed and minimally processed, may be contaminated with pesticide residues, exposing the population to these toxic substances. In this sense, foods considered healthy are most likely to be contaminated with various residues.

Table 1. Number of pesticides allowed in the foods offered on the analyzed menus from the restaurants.

| Restaurants | Number of active pesticide ingredients allowed for use* | Number of food items exposed |
|---|---|---|
| 1 | 269 | 45 |
| 2 | 250 | 31 |
| 3 | 236 | 23 |
| 4 | 281 | 47 |
| 5 | 306 | 56 |
| 6 | 238 | 36 |
| Average | 263 | 40 |
| SD | 27.32 | 11.96 |

* Brazilian pesticide database [4] data from March 2023; SD: standard deviation; RI: institutional restaurant.

## 3.2. The foods offered on the menus and their exposure to pesticides

The AI of products sold in Brazil authorized for use in foods on the menu were identified. An average of 263 active pesticide ingredients were observed for approximately 40 foods. Table 1 presents a general analysis of the number of pesticides allowed in the foods offered on the analyzed menus.

We emphasize that the restaurant menus' foods are similar and indicate the same menu pattern. This information reinforces the premise that we are not evaluating the quality of restaurants but rather the food offered from the perspective of contamination by pesticide residues.

From a general assessment perspective regarding the permission to use pesticides in food production in Brazil, it can be inferred that cereals, legumes, and vegetables are possibly the most exposed to active pesticide ingredients (Fig 2). The foods that stood out regarding the

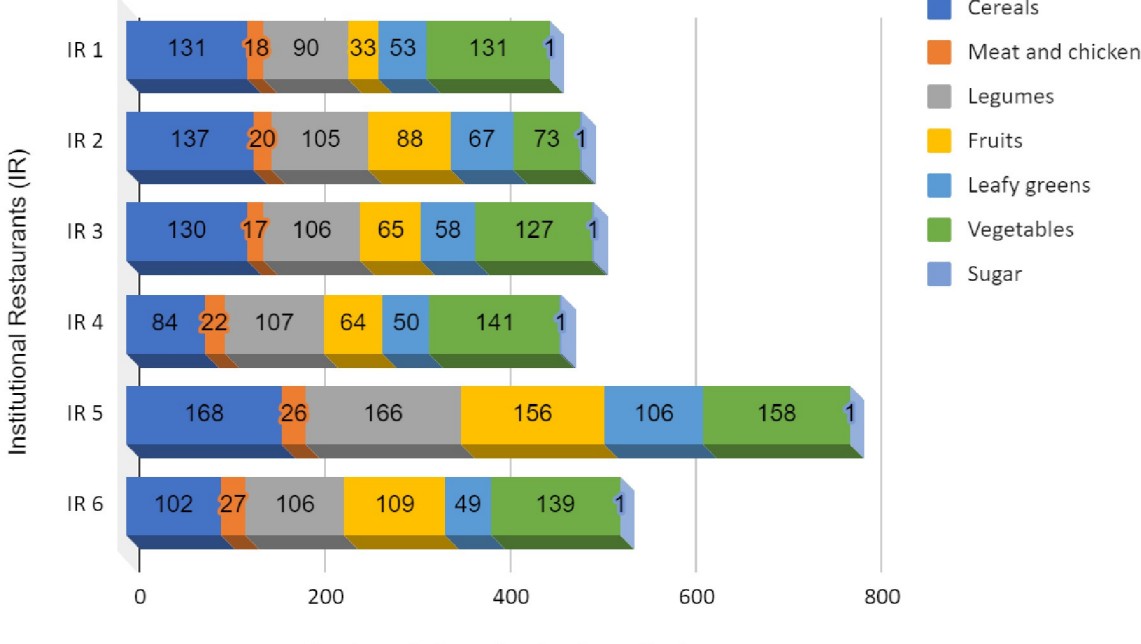

Fig 2. Number of allowed active ingredients for use in foods offered on restaurant menus categorized by food group.

most significant risk of exposure and possible contamination were beans, rice, pasta (wheat), potatoes, carrots, and yams. It is important to note that these foods are commonly consumed in Brazil and are often exposed to pesticide residues through conventional production methods. The data for the IR showed a more significant number of active ingredients since its menus had greater diversity, including fruits, cereals, and legumes, which resulted in a greater quantity of foods from these groups.

This general analysis estimated that the restaurant food purchased from conventional agriculture may receive pesticides throughout the production and supply chain. Data suggests that beans may contain pesticide residues, as observed in a previous study using the QuEChERS method and gas chromatography with an electron capture detector. The authors analyzed 11 types of food and found contamination in beans, being one of the most contaminated, cowpeas (56.8%), black beans (56.5%), and red beans (54.0%), showing positivity in more than half of the analyzes [40]. Additionally, another study detected contamination of beans by lindane, the pesticide with the highest concentration in the samples analyzed [41]. However, these data cannot be equated with data on the possible occurrence of pesticide residues in Brazil, as there are different planting techniques, authorized pesticides, and maximum residue limits (MRL).

Studies to monitor pesticide residues in food, such as the National Program for the Control of Residues and Contaminants in Products of Vegetable Origin [42] analyzed 853 food samples; 150 were non-compliant, emphasizing beans, bell peppers, grapes, and tomatoes. Some of these foods were found on the menus of the evaluated restaurants.

Over the years, it has been discovered that pesticide residues are found not only in plant-based foods but also in animal-based products. Animals ingest these residues through the feed and forage they consume. The contamination is worsened by the persistence of residues in the environment, its lipophilic nature, and the application of veterinary products in animals [43].

The results presented in Fig 2 for the meat and poultry group are very similar. In restaurant 1, 18 ingredients from pesticides are permitted for use, restaurant 2 (20 active ingredients), restaurant 3 (17 active ingredients), restaurant 4 (22 ingredients of pesticides), restaurant 5 (26 active ingredients), and restaurant 6 (27 active ingredients), demonstrating the possible contamination arising from these food groups. To this end, it is important to highlight that in addition to the various active ingredients authorized for this group, the issue of bioaccumulation through lipophilic compounds must also be considered since animals have this capacity. Moreover, the results are very similar, possibly once those foods are daily on the menus combined with a high per capita.

In a study by Dasriya et al. [44] on milk, cereals, and fruit juice samples, pesticide residues were detected using the paper strip sensor method. The results showed that 33 milk samples were considered positive for pesticide residue. Residues of chlorpyrifos and chlorpyrifos-methyl were above the MRL allowed by the Codex Alimentarius Commission.

A narrative review on pesticide residues in animal-source foods shows that many studies have highlighted active ingredient presence in these foods. Carbamates and organophosphates were found in raw milk; organochlorines in cheeses, pasteurized milk, butter, yogurt, eggs, chicken, and beef meat; and organochlorines and polychlorinated biphenyls in fish, mollusks, and crustaceans [45].

When analyzing the findings related to ultra-processed animal-source foods, the Brazilian Institute for Consumer Protection [46] (named IDEC in Brazil) in 2022, analyzed samples of these foods. The institute detected pesticide residues in curd cheese, Calabrian pork sausage, mortadella, sausage, hamburger beef, and chicken nuggets. These results reinforce the contamination issue in ultra-processed foods.

A study by Valentim et al. [47] demonstrates that public programs for pesticides control and monitoring must improve their methodologies to ensure transparency and greater

consumer safety. Moreover, the study highlights that the programs do not adopt multi-exposure risk assessment since reports indicate samples contaminated by more than one active ingredient. The cumulative effects must be considered in methodologies for analyzing pesticide residues in food.

Our study revealed the potential accumulation of a.i contamination in a single food item (Fig 3). The following foods were found to contain multiple pesticide residues: beans, rice, pasta, potatoes, tomatoes, milk, beef, and chicken. The same was also observed in ultra-processed foods such as ketchup, straw potatoes, soy sauce, and cream, which possibly concentrate several types of pesticides in isolation due to the higher amount of active ingredient allowed in their crops, as shown in the data in Fig 3.

Some foods like garlic, bananas, bell peppers, carrots, and pumpkins may contain fewer pesticides. On the other hand, ultra-processed products like ketchup and straw potatoes are often high in active ingredients. The reason could be that these products are made with contaminated ingredients, such as tomatoes and potatoes, which already have pesticide residues. For this analysis, an estimate was made of how much food could contain various pesticide residues. The analysis is based on a careful review of the Brazilian pesticide database and examines the quantity of a.i. allowed for crops. Understanding this information is crucial because it helps us identify the potential negative impact of multiple exposure and the synergistic effect of these compounds on our health. It is important to note that for food crops there are several types of AI allowed and it is uncertain how many are used. When testing for pesticide residues, multiple residues may be found in the same sample due to the application of different types of pesticides used against different pests or diseases.

According to studies on multiple residues in food, about 40 types of pesticides were found in samples of vegetables and fruits analyzed in Sharkia, Egypt. These pesticides belong to different classes, including insecticides, fungicides, and herbicides. The most contaminated foods

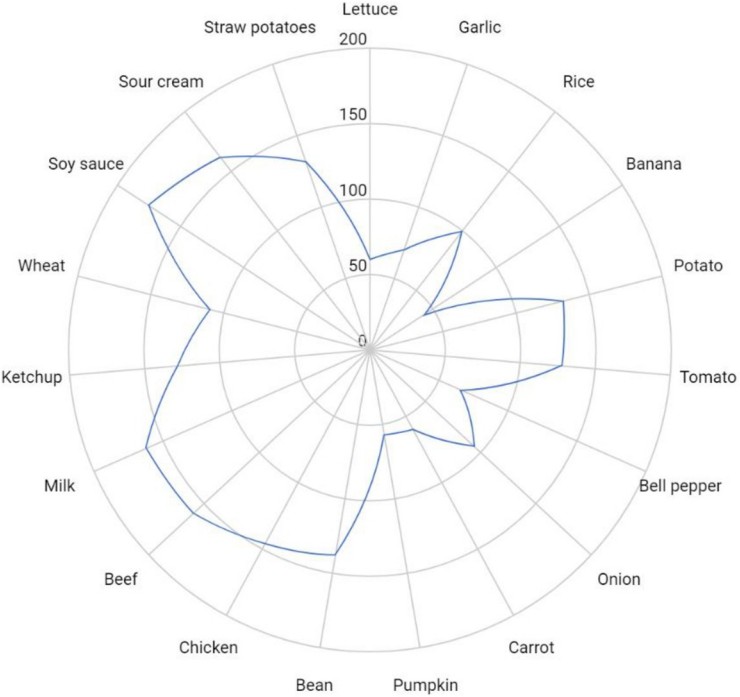

**Fig 3. Occurrence of multiple pesticide residues in foods offered on restaurant menus.**

were cucumbers and apples [48]. Based on a Gas Chromatography-Mass Spectrometry/Mass Spectrometry (GC-MS/MS) detection method for multiple residues, 85.7% of cucumber samples, 83.3% of leeks, and 81.8% of watermelons contained two or more pesticides [49]. The occurrence of multiple pesticide residues in foods such as fruits and vegetables are also observed in other studies conducted in Brazil, China, and Argentina [50–52].

Formulations containing pesticides are complex due to the presence of additives, emulsifiers, and solvents, in addition to the active ingredient. In agriculture, different formulations are commonly used, with varying combinations depending on the crop, making exposure complex and monitoring this multi-exposure difficult to control [53].

The study points to the evidence and mechanisms by which pesticides are linked to chronic diseases, including cancer, neurological disorders, and endocrine disruption. Genotoxicity and proteotoxicity are two mains involved mechanisms [54], via induction of structural or functional damage to chromosomes, DNA, and histone proteins, or indirectly disrupting the profile of genes expression through impairment of cellular organelles like mitochondria and endoplasmic reticulum, nuclear receptors, endocrine network, and the other factors involved in maintenance of cell homeostasis and epigenetic modifications [55]

## 3.3. Characterization of permitted AI

In this study we examined the active ingredient approved for Brazilian food crops (S1 Table). The pesticide was evaluated according to the frequency in which they appeared on the menus evaluated, considering the occurrence of foods. The active ingredients of pesticides were analyzed regarding the toxicological profile of the pesticides [17], agronomic classification [34] and bioaccumulation potential. There was a more significant presence of fungicides and insecticides on the menus. Furthermore, the presence of pesticides classified as highly toxic (fenpyroximate, mandipropamid, methomyl and pirimicarb) and extremely toxic pesticides (paraquat dichloride, metconazole and paraquat) was also observed.

In total, 77 a.i were identified and described, with insecticides being the most common, with 33 a.i. (43%), followed by fungicides, with 31 a.i. (40%), and herbicides, with 11 a.i. (14%). The group of insecticides appears prominently, possibly due to their use from the beginning of planting until post-harvest. Insecticides that possibly appeared on the menus were abamectin, acephate, imidacloprid, lambda-cyhalothrin, and mandipropamid, among others. As for fungicides, they were acetamiprid, azoxystrobin, chlorothalonil, propamocarb, and trifloxystrobin, among others. A highlight for acetamiprid, imidacloprid, acephate, and azoxystrobin is the most detected AI in the latest PARA report [20]. Regarding the 11 herbicides analyzed, it is worth highlighting glyphosate, Brazil's most used active ingredients. Glyphosate possibly appeared 23 times in the evaluated menus and was also irregularly present in samples of wheat, beans, and cassava in the latest PARA report [20].

A study based on epidemiological records of toxicological and individual occurrences investigating pesticide poisoning. Insecticides were the group that appeared the most, accounting for 26% (363 cases), where the predominance is possibly due to the extensive use of insecticides in agriculture and because it is an agent well absorbed through the skin and ingestion [56].

The presence of active ingredients from classes I and II (extremely and highly toxic) was also observed, with three referring to class I and four referring to class II. It is considered that these pesticides appeared possibly with a high frequency on menus, as is the case of mandipropamid, from the highly toxic class, which appeared 25 times on menus. Other examples are methomyl (22 times), fenpyroximate (14 times) and pirimicarb (14 times). Regarding the

extremely toxic group, paraquat dichloride appeared possibly 13 times on the menus, paraquat 16 times, and metconazole 13 times.

Paraquat, which may possibly be present 16 times on menus, is a type of herbicide in the highly toxic class. It can be used in agriculture to control weeds and is generally applied before vegetable cultivation [57].

Metconazole, which probably be present 13 times on menus, is a type of fungicide in the highly toxic class. It is authorized for use on various crops, such as cotton, garlic, peanuts, oats, potatoes, coffee, onions, carrots, barley, beans, watermelon, melon, corn, strawberries, peppers, soybeans, tomatoes, wheat, and grapes [58].

Regarding the results of the bioaccumulation potential of the 77 compounds analyzed, 42 showed high potential. Of these, 31 are present in meat and poultry, possibly due to their high potential to bioaccumulate, as some pesticides are relatively lipophilic and tend to accumulate in the adipose tissues of animals (such as the metabolites of DDT, aldrin, and dieldrin). Some compounds had a possibly high frequency on the menus, such as azoxystrobin (35 times), spiromesifen (31 times), fluxapyroxad (35 times), pyraclostrobin (35 times), trifloxystrobin (37 times), and tebuconazole (40 times).

Regarding tebuconazole, it is a systemic fungicide with a high capacity for bioaccumulation. It is the active ingredient possibly more prevalent on menus and is a type of substance banned in Europe as it can cause changes to the reproductive system and fetal malformations [37]. Also noteworthy is lambda-cyhalothrin, an insecticide with high bioaccumulation capacity and lipophilic properties. Lambda-cyhalothrin possibly appeared 25 times in the analyzed menus and showed irregularities in samples evaluated in the latest Brazilian pesticide monitoring report [20] for apple, grape, cabbage, wheat, broccoli, bell pepper, cucumber, and banana crops.

Trifloxystrobin, which may possibly present 37 times on menus and has a high capacity for bioaccumulation, is associated with changes in the levels of several lipids in neuronal cells, as well as the inhibition of mitochondrial oxidative respiration, in tests carried out in vitro [59].

According to in vivo studies, pyraclostrobin, which possibly appeared 35 times on menus, can be considered as an inducer of high levels of hydrogen peroxide, malondialdehyde, and reactive oxygen species, as well as cellular regulator (p53) genes and decreased apoptotic anti-inflammatory gene expression [60]

Due to the lipophilic ability of pesticides to accumulate in adipose tissue, people with obesity, or those with a high percentage of fat may be more exposed to the risk of these substances with a high potential for bioaccumulation. Pesticides with a high Kow value can be stored in human adipose tissues and the breast, for example, which is an organ with a high fat content. These chemical compounds accumulate and store highly hydrophobic residues, prolonging the elimination rate [61]. Some compounds are also considered environmentally obesogenic, capable of inducing lipid accumulation via PPARγ activation, and considered the key regulator of adipogenesis [62]

From this perspective, the study by Rodrigues [63] investigated the adipogenic potential of the pesticides ametrine and fenbutatin oxide. These pesticides were investigated using the adipocyte differentiation assay, and the effect on PPARγ and RXRα was evaluated using the transfection and luciferase reporter gene assay. As a result, both ametrine and fenbutatin oxide induced lipid accumulation in 3T3-L1 cells, with ametrine causing a more significant accumulation. However, the evidence still needs to be more conclusive, and the subject must be studied and discussed, specifically regarding the mechanisms involved.

Furthermore, about half of the studies considered by Wang et al. [64] were cross-sectional studies in which the disease/exposure may change over time. In this sense, long-term follow-up studies are required to assess the impacts of chemical pollutants on the risk of obesity and dose-response.

### 3.4. Active ingredients (a.i.) of approved pesticides and the corresponding percentage of Acceptable Daily Intake (ADI)

The results of the pesticides a.i. possibly found in the menus according to their ADI contribution are presented in Table 2. Most AI can be found in foods at levels up to 10% of the ADI. This data is alarming, mainly due to the excessive content of different substances in the same food. Even more worrying are those ingredients that showed intake that possibly exceeded the ADI values (>100%). It was the case for some food groups, such as cereals, meat and poultry, legumes, fruits, vegetables, roots and tubers, and sauces. The meat and poultry group presented a higher frequency of ingredient, above 100% of the ADI, and was included on the menus alternately daily.

This result of % ADI above 100% is found in all restaurants evaluated and for the cereals and legumes group. Regarding animal-source foods, the data is concerning. Over 100 AI approved for these foods were analyzed in each restaurant, and the majority had an ADI percentage of up to 10%. Likewise, beans and rice presented several active ingredients in the range of up to 10%. These are foods consumed daily by the Brazilian population.

Rice and beans are essential foods in the Brazilian population's eating habits and represent important sources of nutrients. However, their crops have various authorized pesticides with various indications for use, which may cause concern with this multiple exposure [65].

Regarding the group that possibly had the most significant number of a.i., we can highlight the group of vegetables. In all the restaurants evaluated, this was the group with the highest number of a.i., followed by the meat and poultry group, which in restaurants 1, 3, and 6 were the ones that appeared the most, followed by the fruit group in restaurants 4 and 5, and cereals for restaurant 2.

Table 3 systematized some results to highlight the risk of exposure. There is the possible presence of residues above 100% of the ADI of cyproconazole in rice, abamectin, buprofezin, and fluazifop-p-butyl in beans, carbaryl, dichlorvos and malathion in pasta, aldrin and dieldrin, DDT, endrin, chlordane, haloxyfop and heptachlor in meat and poultry, abamectin in potatoes and carrots, cymoxanil in grape pulp, dimethoate and triazophos in ketchup, linuron in yam, diafenthiuron in bell pepper, carbofuran in orange and triforine and abamectin in melon. We emphasize that the estimated presence of these compounds is possible due to their persistence in the environment.

In all the analyzed restaurants, meat and poultry, rice, and beans contained a.i. that exceeded the ADI by 100%. Furthermore, the active ingredients that have been surpassed are repeated. Another fact that can influence the amount of pesticide residue is the *per capita* amount of food; the higher it is, the greater the amount of exposure to the active ingredient. Another relevant factor is exposure to cyproconazole, possibly exceeding the ADI in rice, a food commonly consumed by Brazilians. This active ingredient may be related to the development of liver cancer, according to a systematic review carried out by Valentim et al. [47]

Another finding of present study was related to the residues possibly present in the melon. Melons hold substantial economic importance, particularly in the Northeast region of Brazil, being one of the most exported fruits [66]. For this food, possibly the ADI exceeded 100% in restaurant 4 and restaurant 5, and orange exceeded the ADI by 100% for carbofuran. There is no data on oranges in the Brazilian pesticide database. However, it presented irregular results in the Brazilian pesticide report for the 2017–2018 cycle and the most recent report for the 2018–2019 and 2022 cycle. Carbofuran was one of the substances most detected in the samples analyzed and is currently banned in Brazil [20]. A study conducted with honeysuckle (*Lonicera japonica* Thunb.) assessed the presence of 93 pesticide residues, including carbamates, pyrethroids, triazoles, neonicotinoids, organophosphorus, organochlorines, and found that 86% of

**Table 2. Percentage of Acceptable Daily Intake (ADI) of active ingredients by food groups offered on the menus of institutional restaurants.**

| RESTAURANT | FOOD GROUPS ON THE MENUS | ACTIVE INGREDIENTS ACCORDING TO ADI RANGE | | | | | | | | | | TOTAL |
|---|---|---|---|---|---|---|---|---|---|---|---|---|
| | | 0,1–10% | 10–20% | 20–30% | 30–40% | 40–50% | 50–60% | 70–80% | 80–90% | 90–100% | >100% | |
| 1 | Cereals | 107 | 3 | 1 | 2 | 0 | 0 | 0 | 0 | 0 | 2 | 115 |
| | Meat and poultry | 206 | 9 | 4 | 2 | 0 | 1 | 0 | 0 | 2 | 7 | 231 |
| | Legumes | 84 | 3 | 0 | 0 | 0 | 0 | 0 | 0 | 0 | 1 | 88 |
| | Fruits | 88 | 6 | 0 | 1 | 0 | 0 | 1 | 0 | 0 | 1 | 97 |
| | Vegetables and greens | 356 | 2 | 0 | 0 | 0 | 0 | 0 | 0 | 0 | 0 | 358 |
| | Sugar and sweets | 0 | 0 | 0 | 0 | 0 | 0 | 0 | 0 | 0 | 0 | 0 |
| | Oil and fat | 11 | 1 | 0 | 0 | 0 | 0 | 0 | 0 | 0 | 0 | 12 |
| | Milk and dairy | 117 | 1 | 0 | 0 | 0 | 0 | 0 | 0 | 0 | 0 | 118 |
| | Eggs | 30 | 1 | 0 | 0 | 0 | 0 | 0 | 0 | 0 | 0 | 31 |
| | Sauces | 174 | 11 | 3 | 4 | 1 | 0 | 0 | 0 | 0 | 2 | 195 |
| | Roots and tubers | 166 | 6 | 3 | 2 | 0 | 1 | 0 | 0 | 0 | 0 | 178 |
| | Spices | 0 | 0 | 0 | 0 | 0 | 0 | 0 | 0 | 0 | 0 | 0 |
| 2 | Cereals | 253 | 6 | 2 | 0 | 0 | 1 | 2 | 0 | 0 | 2 | 266 |
| | Meat and poultry | 205 | 9 | 4 | 2 | 0 | 1 | 0 | 0 | 2 | 7 | 230 |
| | Legumes | 116 | 3 | 1 | 0 | 0 | 0 | 0 | 1 | 1 | 1 | 123 |
| | Fruits | 146 | 9 | 5 | 0 | 1 | 0 | 1 | 0 | 1 | 1 | 164 |
| | Vegetables and greens | 378 | 9 | 2 | 2 | 0 | 0 | 0 | 0 | 0 | 1 | 392 |
| | Sugar and sweets | 1 | 0 | 0 | 0 | 0 | 0 | 0 | 0 | 0 | 0 | 1 |
| | Oil and fat | 41 | 1 | 0 | 0 | 0 | 0 | 0 | 0 | 0 | 0 | 42 |
| | Milk and dairy | 99 | 1 | 0 | 0 | 0 | 0 | 0 | 0 | 0 | 0 | 100 |
| | Eggs | 32 | 0 | 0 | 0 | 0 | 0 | 0 | 0 | 0 | 0 | 32 |
| | Sauces | 102 | 0 | 0 | 0 | 0 | 0 | 0 | 0 | 0 | 0 | 102 |
| | Roots and tubers | 136 | 0 | 0 | 2 | 0 | 1 | 0 | 0 | 0 | 0 | 139 |
| | Spices | 0 | 0 | 0 | 0 | 0 | 0 | 0 | 0 | 0 | 0 | 0 |
| 3 | Cereals | 184 | 6 | 2 | 1 | 0 | 1 | 1 | 0 | 0 | 4 | 199 |
| | Meat and poultry | 207 | 6 | 5 | 1 | 1 | 0 | 2 | 0 | 0 | 7 | 229 |
| | Legumes | 114 | 6 | 1 | 0 | 0 | 0 | 0 | 0 | 0 | 3 | 124 |
| | Fruits | 127 | 3 | 1 | 2 | 0 | 0 | 0 | 0 | 0 | 0 | 133 |
| | Vegetables and greens | 326 | 4 | 0 | 1 | 0 | 0 | 0 | 0 | 0 | 0 | 331 |
| | Sugar and sweets | 1 | 0 | 0 | 0 | 0 | 0 | 0 | 0 | 0 | 0 | 1 |
| | Oil and fat | 0 | 0 | 0 | 0 | 0 | 0 | 0 | 0 | 0 | 0 | 0 |
| | Milk and dairy | 42 | 0 | 0 | 0 | 0 | 0 | 0 | 0 | 0 | 0 | 42 |
| | Eggs | 0 | 0 | 0 | 0 | 0 | 0 | 0 | 0 | 0 | 0 | 0 |
| | Sauces | 102 | 0 | 0 | 0 | 0 | 0 | 0 | 0 | 0 | 0 | 102 |
| | Roots and tubers | 52 | 0 | 0 | 1 | 0 | 0 | 0 | 0 | 0 | 0 | 53 |
| | Spices | 0 | 0 | 0 | 0 | 0 | 0 | 0 | 0 | 0 | 0 | 0 |
| 4 | Cereals | 159 | 6 | 2 | 0 | 0 | 0 | 0 | 1 | 0 | 2 | 170 |
| | Meat and poultry | 220 | 10 | 6 | 1 | 1 | 2 | 1 | 0 | 0 | 8 | 249 |
| | Legumes | 117 | 4 | 2 | 1 | 0 | 0 | 0 | 0 | 0 | 3 | 127 |
| | Fruits | 304 | 15 | 7 | 4 | 1 | 1 | 0 | 0 | 0 | 3 | 335 |
| | Vegetables and greens | 536 | 10 | 0 | 1 | 0 | 0 | 0 | 0 | 1 | 0 | 548 |
| | Sugar and sweets | 94 | 3 | 1 | 1 | 0 | 0 | 0 | 0 | 0 | 0 | 99 |
| | Oil and fat | 6 | 0 | 0 | 0 | 0 | 0 | 1 | 1 | 0 | 0 | 8 |
| | Milk and dairy | 90 | 1 | 0 | 0 | 0 | 0 | 0 | 0 | 0 | 0 | 91 |
| | Eggs | 30 | 1 | 0 | 0 | 0 | 0 | 0 | 0 | 0 | 0 | 31 |
| | Sauces | 322 | 0 | 0 | 0 | 0 | 0 | 0 | 0 | 0 | 0 | 322 |
| | Roots and tubers | 133 | 1 | 0 | 1 | 1 | 0 | 0 | 0 | 0 | 0 | 136 |
| | Spices | 4 | 0 | 0 | 0 | 0 | 0 | 0 | 0 | 0 | 0 | 4 |

*(Continued)*

**Table 2.** (Continued)

| RESTAURANT | FOOD GROUPS ON THE MENUS | ACTIVE INGREDIENTS ACCORDING TO ADI RANGE | | | | | | | | | | TOTAL |
|---|---|---|---|---|---|---|---|---|---|---|---|---|
| | | 0,1–10% | 10–20% | 20–30% | 30–40% | 40–50% | 50–60% | 70–80% | 80–90% | 90–100% | >100% | |
| 5 | Cereals | 241 | 5 | 2 | 1 | 0 | 2 | 0 | 0 | 0 | 4 | 255 |
| | Meat and poultry | 212 | 9 | 5 | 4 | 2 | 1 | 0 | 0 | 0 | 9 | 242 |
| | Legumes | 295 | 5 | 2 | 1 | 0 | 0 | 0 | 1 | 1 | 1 | 306 |
| | Fruits | 462 | 28 | 10 | 3 | 3 | 2 | 0 | 1 | 1 | 2 | 512 |
| | Vegetables and greens | 656 | 10 | 4 | 1 | 0 | 0 | 1 | 0 | 0 | 1 | 673 |
| | Sugar and sweets | 63 | 1 | 0 | 0 | 0 | 0 | 0 | 0 | 0 | 0 | 64 |
| | Oil and fat | 68 | 1 | 0 | 1 | 0 | 0 | 2 | 0 | 0 | 0 | 72 |
| | Milk and dairy | 249 | 2 | 1 | 0 | 0 | 0 | 0 | 0 | 0 | 0 | 252 |
| | Eggs | 0 | 0 | 0 | 0 | 0 | 0 | 0 | 0 | 0 | 0 | 0 |
| | Sauces | 485 | 10 | 4 | 4 | 0 | 0 | 0 | 0 | 0 | 0 | 503 |
| | Roots and tubers | 237 | 9 | 3 | 1 | 1 | 1 | 0 | 0 | 0 | 2 | 254 |
| | Spices | 0 | 0 | 0 | 0 | 0 | 0 | 0 | 0 | 0 | 0 | 0 |
| 6 | Cereals | 107 | 5 | 2 | 0 | 0 | 1 | 0 | 0 | 0 | 4 | 119 |
| | Meat and poultry | 219 | 8 | 7 | 3 | 2 | 2 | 0 | 0 | 0 | 9 | 250 |
| | Legumes | 122 | 1 | 1 | 0 | 0 | 0 | 0 | 1 | 1 | 1 | 127 |
| | Fruits | 180 | 10 | 2 | 2 | 0 | 1 | 1 | 1 | 0 | 0 | 197 |
| | Vegetables and greens | 280 | 4 | 0 | 0 | 0 | 0 | 0 | 0 | 0 | 0 | 284 |
| | Sugar and sweets | 63 | 1 | 0 | 0 | 0 | 0 | 0 | 0 | 0 | 0 | 64 |
| | Oil and fat | 40 | 1 | 0 | 0 | 0 | 0 | 0 | 0 | 0 | 0 | 41 |
| | Milk and dairy | 146 | 1 | 0 | 0 | 0 | 0 | 0 | 0 | 0 | 0 | 147 |
| | Eggs | 30 | 1 | 0 | 0 | 0 | 0 | 0 | 0 | 0 | 0 | 31 |
| | Sauces | 0 | 0 | 0 | 0 | 0 | 0 | 0 | 0 | 0 | 0 | 0 |
| | Roots and tubers | 119 | 6 | 1 | 0 | 0 | 0 | 1 | 0 | 0 | 0 | 127 |
| | Spices | 0 | 0 | 0 | 0 | 0 | 0 | 0 | 0 | 0 | 0 | 0 |

Supplementary database available in S2 Table

the samples were contaminated with at least one pesticide. Surprisingly, the banned pesticide carbofuran was also identified [67].

In the menus, we have the offer of grape juice, which also possibly exceeded the ADI by 100%, there's an important consideration to explore alternative options. One potential strategy

**Table 3. Active ingredients that exceeded the Acceptable Daily Intake (ADI) values and the respective foods exposed.**

| Restaurant | Pesticides residue (n) | Active ingredients | Food exposed |
|---|---|---|---|
| 1 | 7 | Cyproconazole, dimethoate, triazophos, haloxyfop, heptachlor, chlordane, cymoxanil | Rice, ketchup, meat, poultry, grape pulp |
| 2 | 8 | Abamectin, carbaryl, dichlorvos, malathion, cymoxanil, haloxyfop, heptachlor, chlordane | Potatoes, rice, beans, pasta, grape pulp, meat and poultry |
| 3 | 9 | Abamectin, buprofezin, carbaryl, chlordane, dichlorvos, fluasifop-p-butyl, haloxyfop, heptachlor, malathion | Rice, poultry, meat, beans, pasta |
| 4 | 10 | Abamectin, buprofezin, carbofuran, chlordane, endrin, haloxyfop, heptachlor, triforin, fluasifop-p-butyl, cymoxanil | Poultry, meat, melon, orange, beans, rice, grape pulp. |
| 5 | 11 | chlordane, endrin, haloxyfop, heptachlor, cymoxanil, carbaryl, dichlorvos, malathion, abamectin, diafentiuron, linuron | Meat, poultry, grape pulp, pasta, beans, rice, peppers, melon, yams, carrots |
| 6 | 8 | Abamectin, carbaryl, chlordane, endrin, dichlorvos, haloxyfop, heptachlor, malathion | Rice, poultry, meat, beans, pasta |

could be to switch to fruit juice sourced from the region, as it may contain lower pesticide levels compared to the grape pulp.

Regarding the active ingredient DDT, banned since the 1970s but still present in meat and poultry, is a type of organochlorine compound that is highly persistent in the environment. A bibliographical survey study by Menck et al. [68] analyzed the concentrations of pesticides in human milk, with DDE and DDT being the ones that appeared most in the studies consulted.

The adipose tissue is the primary receptor for many insecticides. Therefore, these types of pesticides accumulate in fats [69]. DDT has been banned for years, but its persistence in the environment is still noticeable nowadays, as is the active ingredient Aldrin. According to the International Agency for Research on Cancer [70], they are considered human carcinogens and are included in group B1, having been associated with the development of liver cancer, respiratory tract cancer, and lymphomas [54].

Malathion, possibly present in pasta, has low toxicity for mammals and relatively high toxicity for fish. Its contamination can occur through direct application, drift, aerial spraying, and atmospheric leaching through precipitation, erosion, and runoff from agricultural lands [71]. Like the active ingredient carbaryl, which was also possibly present in the noodles, it has been widely used in agriculture due to its broad-spectrum effectiveness in controlling more than 100 species of insects. However, it is a dangerous product for consumers' health and the environment [72].

With respect to the calculation of %IDA, this is a safety parameter defined as the maximum amount of pesticide that can be ingested daily throughout one's life so as not to cause harm to health. However, the calculation is based on a weight of 60 kg, which does not consider older adults and children who do not reach this weight and are more vulnerable to risks.

The literature reinforces that the ADI calculation for the entire population is based on an average value, which does not consider individual characteristics, behavior, genetics, and physiological functioning. The risk assessment is based on studies carried out from the exposure of a single compound, not considering all the health effects of multiple exposures and synergism of different substances.

It is important to note that being within the ADI range does not necessarily mean that a food is safe to consume. It is crucial to consider that consuming foods containing pesticide residues is daily, and diets consist of various foods, not just one. Therefore, a meal can quickly become a mixture of pesticide residues. Combining these compounds can cause chemical interactions in the body, as most studies analyze the active ingredients in isolation, not considering dose studies on the analysis of chronic and cumulative risk. When this % IDA exceeds 100%, it is even more worrying, as is the case with the presentation of the results in Table 3. Moreover, MRL safety regulations are often violated, resulting in exceedance of the allowable limit. This can lead to worsening health conditions in the population.

A study similar to the present article was conducted by Meira and Silva [73]. This study evaluates the Theoretical Maximum Daily Intake (TMDI) of pesticides potentially present in the regular diet of students (n = 341) enrolled in public schools in Brazil, comparing these values with the Acceptable Daily Intake (ADI) set by regulatory authorities. The results showed that the median estimated intake of nine of the 272 pesticides potentially present in the students' diet exceeded the ADI levels established by ANVISA. Moreover, the maximum intake of 58 pesticides surpassed ANVISA's limits, highlighting the urgency of reducing the levels of these substances in common foods of the Brazilian diet, especially given the health risks to children.

These limits are seen to be extended in the most recent report, which was released in 2023. In the 2018–2019 cycle, 25.6% of the samples analyzed presented nonconformities, with 2.4% above the permitted limit. In the 2022 cycle, 25% of non-conformities were found, 4% above

the permitted limit [20]. It is worth reinforcing the limitations regarding the MRL since, in Brazil, the limit allowed in crops is much higher than that allowed in other countries.

These data can be identified in a survey carried out by Bombardi [74], where for the active ingredient glyphosate, its limit allowed in water in Brazil is 500 ug/L, while in the European Union (0.1 ug/L), that is, 5,000 times greater in Brazil. Chlorothalonil in lettuce cultivation is allowed in Brazil (6 mg/kg) at a level 600 times higher than the European Union's limit (0.01 mg/kg).

In this way, Brazil diverges from other countries since there is no safe dose. In addition to the precariousness of residue monitoring in Brazil compared to other countries, the country's use of pesticides and the limits allowed in crops is still very flexible. In addition, active ingredients such as carbendazim, chlorpyrifos, and acephate, banned in the European Union, are widespread in the national territory [75].

## 3.5. Estimate of Theoretical Maximum Daily Intake (TMDI) and risk of exposure to pesticides in foods offered on menus

The analysis to determine the amount of pesticide possibly intake was carried out using anthropometric data on body weight. This data was obtained from the sample conducted by Brazilian family budget program [38], where the average weight of the population was 65 kg, considering individuals between the ages of 18 and 60. Specific criteria were established to evaluate exposure and assess risk in 120 menus. Twelve active ingredients with high occurrence/frequency, toxicity, and detection were listed based on the 2017 and 2018 Brazilian reports [20]. The active ingredients were evaluated in a daily menu consisting of a traditional lunch dish, which includes meat or chicken, rice, beans, salad and/or vegetables, and fruit and/ or juice, as most foods are similar (Table 4).

The TMDI was higher than the ADI (% ADI > 100) in at least one restaurant. In the analysis of daily restaurant menus, we found that the active ingredient methomyl is possibly found in quantities above the ADI, characterizing its intake as an unacceptable risk. The remaining active ingredients had an estimated average daily intake within ADI. However, we highlight a concern about cumulative exposure over time, given that food consumption is daily and several times a day.

Methomyl is an insecticide that can be used both by direct contact and as a systemic treatment. It belongs to the methylcarbamate chemical group. It is currently authorized for use in soybean, corn, wheat, beans, rice, potatoes, tomatoes, pineapple, acerola, melon, and watermelon crops. Its toxicological classification for acute effects is category II, highly toxic PARA's latest report showed 40 irregular detections in cucumber samples (17.24%) [20].

Regarding the estimated weekly cumulative risk, we observed that other active ingredients, besides methomyl, could be above the ADI. This was the case with acephate, terbufos, bifenthrin, carbaryl, and chlorothalonil, representing an unacceptable risk for consumers. When analyzing rice and beans, the basis of the Brazilian diet, we highlighted the possibility of 186 pesticide residues in the sum and combination of these foods, offering possible risks to the consumer. Furthermore, from a weekly analysis perspective considering the complete lunch meal, we found, for example, an unacceptable risk for ingesting acephate.

Friedrich et al. [75] report that acephate is one of the pesticides banned by the European community due to its potential endocrine-disrupting effect on humans. Furthermore, the active ingredient had the most irregular detections in the latest PARA report, where of the 1,772 samples, 90 showed irregularities, including active ingredients not allowed for cultures. Data also referring to the results of the Brazilian report cycle 2018–2019 and 2022, an irregularity was found for acephate in samples of papaya (4.05%), zucchini (3.41%), grapes (0.42%),

**Table 4. Assessment of estimated theoretical maximum daily intake (TMDI) and risk of exposure to pesticide residues in foods offered at lunch in restaurants.** Supplementary database available in S2 Table.

| RESTAURANT | Foods comprising a daily lunch menu | Active ingredients | TMDI average (mg/Kg/day) | ADI (mg/kg body weight)* | Risk | TMDI estimative (mg/Kg/weekly) | Estimated cumulative risk** |
|---|---|---|---|---|---|---|---|
| 1 | Meat, Rice, Beans, Salad (Lettuce, Chard, Cabbage, Tomato), Fruit (Pineapple), Juice (Guava) | 2,4 D | 0,032 | 0,650 | A | 0,162 | A |
| | | Acephate | 0,021 | 0,078 | A | 0,438 | U |
| | | Bifenthrin | 0,259 | 1,300 | A | 1,297 | A |
| | | Carbaryl | 0,029 | 0,195 | A | 0,147 | A |
| | | Methomyl | 0,038 | 1,300 | A | 0,191 | A |
| | | Cypermethrin | 0,179 | 3,250 | A | 0,895 | A |
| | | Chlorothalonil | 0,295 | 1,950 | A | 1,475 | A |
| | | Chlorpyrifos | 0,038 | 0,650 | A | 0,189 | A |
| | | Glyphosate | 0,026 | 32,500 | A | 0,130 | A |
| | | Imidacloprid | 0,068 | 3,250 | A | 0,339 | A |
| | | Tebuconazole | 0,104 | 1,950 | A | 0,519 | A |
| | | Terbufos | 0,005 | 0,013 | A | 0,027 | U |
| 2 | Poultry, Pasta, Beans, Vegetables (potatoes, carrots, chayote), Fruit (banana), Juice (acerola) | 2,4 D | 0,043 | 0,650 | A | 0,217 | A |
| | | Acephate | 0,008 | 0,078 | A | 0,039 | A |
| | | Bifenthrin | 0,282 | 1,300 | A | 1,409 | U |
| | | Carbaryl | 0,051 | 0,195 | A | 0,255 | U |
| | | Methomyl | 1,330 | 1,300 | U | 6,652 | U |
| | | Cypermethrin | 0,219 | 3,250 | A | 1,096 | A |
| | | Chlorothalonil | 0,267 | 1,950 | A | 1,333 | A |
| | | Chlorpyrifos | 0,030 | 0,650 | A | 0,151 | A |
| | | Glyphosate | 0,029 | 32,500 | A | 0,145 | A |
| | | Imidacloprid | 0,068 | 3,250 | A | 0,339 | A |
| | | Tebuconazole | 0,143 | 1,950 | A | 0,714 | A |
| | | Terbufos | 0,005 | 0,013 | A | 0,027 | U |
| 3 | Poultry, Rice, Beans, Salad (cucumber, tomato, lettuce, onion), Fruit (Papaya) | 2,4 D | 0,036 | 0,650 | A | 0,178 | A |
| | | Acephate | 0,004 | 0,078 | A | 0,020 | A |
| | | Bifenthrin | 0,212 | 1,300 | A | 1,060 | A |
| | | Carbaryl | 0,046 | 0,195 | A | 0,233 | U |
| | | Methomyl | 0,060 | 1,300 | A | 0,299 | A |
| | | Cypermethrin | 0,138 | 3,250 | A | 0,691 | A |
| | | Chlorothalonil | 0,339 | 1,950 | A | 1,694 | A |
| | | Chlorpyrifos | 0,012 | 0,650 | A | 0,059 | A |
| | | Glyphosate | 0,023 | 32,500 | A | 0,114 | A |
| | | Imidacloprid | 0,037 | 3,250 | A | 0,183 | A |
| | | Tebuconazole | 0,115 | 1,950 | A | 0,575 | A |
| | | Terbufos | 0,007 | 0,013 | A | 0,034 | U |
| 4 | Meat, Rice, Beans, Vegetables (Cassava, pumpkin, carrot, beetroot), Fruit (Melon), Sweet (Guava), Juice (Mango) | 2,4 D | 0,029 | 0,650 | A | 0,147 | A |
| | | Acephate | 0,008 | 0,078 | A | 0,039 | A |
| | | Bifenthrin | 0,272 | 1,300 | A | 1,361 | U |
| | | Carbaryl | 0,055 | 0,195 | A | 0,273 | U |
| | | Methomyl | 0,055 | 1,300 | A | 0,275 | A |
| | | Cypermethrin | 0,272 | 3,250 | A | 1,361 | A |
| | | Chlorothalonil | 0,375 | 1,950 | A | 1,873 | A |
| | | Chlorpyrifos | 0,029 | 0,650 | A | 0,145 | A |
| | | Glyphosate | 0,051 | 32,500 | A | 0,255 | A |
| | | Imidacloprid | 0,082 | 3,250 | A | 0,408 | A |
| | | Tebuconazole | 0,215 | 1,950 | A | 1,075 | A |
| | | Terbufos | 0,009 | 0,013 | A | 0,045 | U |

(*Continued*)

**Table 4.** (Continued)

| RESTAURANT | Foods comprising a daily lunch menu | Active ingredients | TMDI average (mg/Kg/day) | ADI (mg/kg body weight)* | Risk | TMDI estimative (mg/Kg/weekly) | Estimated cumulative risk** |
|---|---|---|---|---|---|---|---|
| 5 | Poultry, Rice, Beans, Salad (Tomato, lettuce, onion, green beans), Fruit (Watermelon), Juice (Cashew) | 2,4 D | 0,035 | 0,650 | A | 0,175 | A |
| | | Acephate | 0,008 | 0,078 | A | 0,043 | A |
| | | Bifenthrin | 0,332 | 1,300 | A | 1,658 | U |
| | | Carbaryl | 0,061 | 0,195 | A | 0,304 | U |
| | | Methomyl | 0,064 | 1,300 | A | 0,321 | A |
| | | Cypermethrin | 0,273 | 3,250 | A | 1,364 | A |
| | | Chlorothalonil | 0,675 | 1,950 | A | 3.377 | U |
| | | Chlorpyrifos | 0,042 | 0,650 | A | 0,210 | A |
| | | Glyphosate | 0,034 | 32,500 | A | 0,168 | A |
| | | Imidacloprid | 0,134 | 3,250 | A | 0,672 | A |
| | | Tebuconazole | 0,256 | 1,950 | A | 1,277 | A |
| | | Terbufos | 0,007 | 0,013 | A | 0,036 | U |
| 6 | Meat, Rice, Beans, Vegetables (potatoes, beets, carrots, chayote, corn), Fruit (Apple), Sweets (Dulce de leche), Juice (guava pulp) | 2,4 D | 0,086 | 0,650 | A | 0,429 | A |
| | | Acephate | 0,006 | 0,078 | A | 0,032 | A |
| | | Bifenthrin | 0,322 | 1,300 | A | 1,610 | U |
| | | Carbaryl | 0,090 | 0,195 | A | 0,452 | U |
| | | Methomyl | 0,039 | 1,300 | A | 0,195 | A |
| | | Cypermethrin | 0,278 | 3,250 | A | 1,391 | A |
| | | Chlorothalonil | 0,356 | 1,950 | A | 1,781 | A |
| | | Chlorpyrifos | 0,028 | 0,650 | A | 0,140 | A |
| | | Glyphosate | 0,030 | 32,500 | A | 0,149 | A |
| | | Imidacloprid | 0,060 | 3,250 | A | 0,300 | A |
| | | Tebuconazole | 0,142 | 1,950 | A | 0,709 | A |
| | | Terbufos | 0,008 | 0,013 | A | 0,038 | U |

A: Acceptable; U: Unacceptable

*Considering body weight of 65 kg

**Estimated cumulative risk was obtained from the sum of TMDI for each AI from Monday to Friday according to the weekly lunch menu.

cabbage (1.14%), cucumber (11.64%), onion (0.78%), passion fruit (27.03%), broccoli (2.80%), cabbage (5.56%), bell pepper (24.65%), beans (2.00%) and potatoes (0.65%) [20].

A recent systematic review conducted by Hess et al. [76] suggests that exposure to acephate can have harmful effects on human health, including damage to sperm, development of type 2 diabetes, hyperglycemia, lipid metabolism dysfunction, DNA damage, and cancer. Moreover, international cancer research [70] has associated acephate exposure with an increased risk of leukemias, non-Hodgkin lymphomas, and pancreatic cancer.

The data from this research warns of the active ingredients bifenthrin, carbaryl, and chlorothalonil, which are likely to pose an unacceptable risk to consumers since these compounds are allowed for use in beans. As for rice, active ingredients such as bifenthrin and chlorothalonil are allowed and this residue is classified as a possible carcinogen for humans (group 2B) [70]

Bifenthrin is an insecticide that has already been detected irregularities in samples of apple (1.57%), pear (11.68%), cabbage (1.14%), zucchini (0.98%), grape (0.42%), coffee (0.63%), orange (7.69%), passion fruit (10.81%), strawberry (13.10%), bell pepper (0.70%) and peanut (1.98%). Additionally, 41 samples were above the maximum allowable residue limit in the 2022 cycle. Effects associated with bifenthrin exposure may be associated with neurotoxicity, obesity, and endocrine disruption [77–79].

Terbufos was also an active ingredient that could be causing an unacceptable risk once its intake was above ADI lunch for a week. Terbufos is an insecticide from the chemical group of organophosphates authorized for banana, sugar cane, beans, corn, and soybean crops.

The insecticide and growth regulator carbaryl is allowed for pineapple, pumpkin, garlic, cotton, banana, potato, onion, cauliflower, beans, apple, cabbage, and tomato crops. Its effects can cause developmental and reproductive toxicity, neurodevelopmental disorders, and impact the immune system, possibly leading to human carcinogenesis [80].

According to the TMDI calculation, the analysis of estimated pesticide intake revealed that the food provided in institutional environments, and which are commonly consumed foods by the Brazilian population, is potentially contaminated with pesticide residues from the entire production chain. Methomyl is the only exception that possibly exceeded the ADI limit in the lunch meal. It is advisable to emphasize that, according to Caldas & Souza [81], risk assessment methodology based on TMDI calculation is conservative, as it assumes that the food supply is always treated with all the registered pesticides for that crop and that one always consumes food containing residues at the tolerance level.

In the food services sector, the cooking stages of food are inherent to processing, and many questions and gaps have not yet been clarified in science regarding whether pesticide residues decrease from food after these processes. A study by Sandlar et al. [82] reaffirms that research on the presence of pesticide residues in food, their reduction during cooking processes, and the associated health risks to humans is still limited. In their study, they assessed the impact of thermal cooking on pesticide residues in seaweed and found that residues decreased after multiple water changes, boiling, and steam cooking. The most prevalent analyzed residues were endrin, DDT, endosulfan, and cypermethrin, highlighting their highly bio accumulative capacity.

We can say that there are no safe levels of pesticides we can consume. We must consider that we consume various foods daily, exposing us to already confirmed evidence of potential health damage. Moreover, the effects of the toxic substances used in pesticides are not fully understood and may have cumulative and synergistic effects on health. Ferrier et al. [83] assert that techniques employed in assessing consumer exposure to pesticides have been consistently reviewed in the European Union and the United States. Several factors impede progress, such as the lack of sufficient data for quantitative analysis, interpretation of probabilistic models, and political and economic obstacles. Particularly in Brazil, this assessment is crucial, considering the cumulative evaluation of pesticide residues. We emphasize that communicating data on chronic exposure to both risk managers and consumers is a significant challenge.

Furthermore, experimental studies in laboratories, which evaluate the potential of a pesticide to generate health problems through its effects on animals, are carried out in a very short time, just three months of testing. The results of this type of study may not necessarily correspond to those found in humans [52]. It is highlighted that the ADI value depends on a series of factors and is not constant but rather a guide used to calculate the permissible limits of these chemicals incorporated into foods. It should consider consumption data, which varies significantly among countries, consumer groups, types, and quantities of consumed diets, making ADI a time-integrated value that should be constantly reassessed [84].

We emphasize in the analysis conducted by Harris et al. [85], the risk assessment remains a simplistic numerical comparison when compared to the hazard assessment based on toxicity studies, however, the intake methodology continues to evolve, allowing quantitative estimates of intake for the entire population, consumers may be able to use risk-benefit analysis to make informed decisions about the risk of pesticide residues in the future.

The chronic effects of pesticide consumption have already been discussed in some studies, and their negative impacts on human health in the long term, mainly relating their

consumption/exposure to the risk of developing various types of cancer. Studies carried out in Brazil have already carried out a multiscale assessment of the main contaminants contained in official reports and the estimate of developing some type of cancer resulting from this exposure, knowing that cancer is a disease that develops in the long term. Some studies highlight the cumulative effects of consuming contaminated food, associated with the presence of pesticide residues above the limit allowed by the WHO, which can impact human health throughout life [47, 86]

The present study had some limitations. Data are estimates from a database that may not reflect the pesticide levels in foods from the region where the institutional restaurants are located. It is important to encourage laboratory analyses to build more consistent databases, considering the different regions of Brazil and the foods consumed in each region. There is no MRL data for all AI, and some foods are absent from the databases, mainly processed and ultra-processed foods; in this sense, the data are estimations of pesticide residues. Moreover, the daily diet was not analyzed; it only included part of the day's usual intake. Due to the absence of anthropometric data on the restaurant consumer population, the average body weight used in the calculations may not have reliably reflected the public served in the restaurants. Furthermore, considering the IR are from educational institutions, the age range of consumers would also include teenagers; thus, the weight used may be overestimated.

## 4. Conclusion

The results of this study indicate a potential contamination of food served in institutional restaurants with pesticide residues, highlighting the risks of chronic exposure. The active ingredient methomyl exceeded the ADI only in the lunch meal. The assessment of pesticide intake estimates in ready-to-consume foods (meals), especially in institutional settings, should be encouraged to promote food safety. Our study makes some recommendations for future research and policy, such as implementing more monitoring programs to regularly of food samples for pesticide residues; promote laboratory analysis; Review and update regulatory standards (limits in food); support sustainable agriculture and educate consumers about the potential risks of pesticide residues.

## Supporting information

**S1 Table. Supplementary material.**
(DOCX)

**S2 Table. Database pesticide project.**
(XLSX)

**S1 File.**
(ZIP)

## Author Contributions

**Conceptualization:** Larissa Mont'Alverne Jucá Seabra, Luciléia Granhen Tavares Colares, Priscilla Moura Rolim.

**Data curation:** Thuany Matias da Silva, Barbara Lettyccya Pereira Chacon de Araújo, Vanessa Cristina da Costa Pires.

**Formal analysis:** Priscilla Moura Rolim.

**Supervision:** Priscilla Moura Rolim.

**Validation:** Priscilla Moura Rolim.

**Writing – original draft:** Thuany Matias da Silva.

**Writing – review & editing:** Larissa Mont'Alverne Jucá Seabra, Luciléia Granhen Tavares Colares, Priscilla Moura Rolim.

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
