## [Decision Letter · Decision Letter 0]

14 Jun 2024

PONE-D-24-15176Food: source of life or disease? Risk assessment of pesticide residue ingestion in food offered by institutional restaurantsPLOS ONE

Dear Dr. ROLIM,

Thank you for submitting your manuscript to PLOS ONE. After careful consideration, we feel that it has merit but does not fully meet PLOS ONE’s publication criteria as it currently stands. Therefore, we invite you to submit a revised version of the manuscript that addresses the points raised during the review process.

**ACADEMIC EDITOR: **Dear Dr. PRISCILLA ROLIM Manuscript ID and Title: PONE-D-24-15176 & Food: source of life or disease? Risk assessment of pesticide residue ingestion in food offered by institutional restaurants I hope this letter finds you well. I am writing to inform you of the editorial decision for your above-said manuscript submitted to the Plos One We have received three Reviewer’s comments on your manuscript, and I would like to thank you for your patience during the review process. I understand that it can be challenging when reviewers provide differing opinions on a submission. Here is a summary of the reviewers' comments: Reviewers # 2 and 3 (Major Revision): found some merits in the research and raised some major points/critics/suggestions (Please find the Reviewer comments) Reviewer #1 has not recommended your manuscript and Rejected by raising some major comments (comments attached) After careful consideration of all three reviewers' feedback and an evaluation of the manuscript by the editorial team, I have recommended your article “Major Revision” following the recommendations provided by Reviewers # 2 & 3. I believe that the comments and suggestions made by Reviewer #1 are comprehensive, and constructive and can significantly enhance the quality of your work in the future. “I strongly recommend to incorporate the suggestions provided by Reviewer #1 with careful consideration and providing suitable rebuttals. Reviewer #1 comments and responses will be critically evaluated during the review process."

We look forward to receiving your revised manuscript.

Kind regards,

Ulaganathan Arisekar

Academic Editor

PLOS ONE

Additional Editor Comments:

Dear Dr. PRISCILLA ROLIM

Manuscript ID and Title: PONE-D-24-15176 & Food: source of life or disease? Risk assessment of pesticide residue ingestion in food offered by institutional restaurants

I hope this letter finds you well. I am writing to inform you of the editorial decision for your above-said manuscript submitted to the Plos One

We have received three Reviewer’s comments on your manuscript, and I would like to thank you for your patience during the review process. I understand that it can be challenging when reviewers provide differing opinions on a submission. Here is a summary of the reviewers' comments:

Reviewers # 2 and 3 (Major Revision): found some merits in the research and raised some major points/critics/suggestions (Please find the Reviewer comments)

Reviewer #1 has not recommended your manuscript and Rejected by raising some major comments (comments attached or find below)

After careful consideration of all three reviewers' feedback and an evaluation of the manuscript by the editorial team, I have recommended your article “Major Revision” following the recommendations provided by Reviewers # 2 & 3. I believe that the comments and suggestions made by Reviewer #1 are comprehensive, and constructive and can significantly enhance the quality of your work in the future.

“I strongly recommend to incorporate the suggestions provided by Reviewer #1 with careful consideration and providing suitable rebuttals. Reviewer #1 comments and responses will be critically evaluated during the review process."

Reviewers' comments:

Reviewer's Responses to Questions

**Comments to the Author**

1. Is the manuscript technically sound, and do the data support the conclusions?

Reviewer #1: No

Reviewer #2: Partly

Reviewer #3: Partly

2. Has the statistical analysis been performed appropriately and rigorously? 

Reviewer #1: N/A

Reviewer #2: Yes

Reviewer #3: Yes

3. Have the authors made all data underlying the findings in their manuscript fully available?

Reviewer #1: No

Reviewer #2: No

Reviewer #3: No

4. Is the manuscript presented in an intelligible fashion and written in standard English?

Reviewer #1: Yes

Reviewer #2: Yes

Reviewer #3: No

5. Review Comments to the Author

Reviewer #1: According to the title and abstract, the manuscript is expected to offer pesticide residue levels in food and the associated risk to consumers in restaurants from educational institutions. But it seems that the levels of pesticide residues have not been analyzed, and the authors have assumed that the food items will contain the MRL levels for all AI with MRL for that food, which is totally unrealistic. Therefore, the results are irrelevant. There are other methodological limitations, for example for pesticides, with toxicokinetic data available, the use of Kow as proxi for bioaccumulation has no sense, as metabolisms, which is very relevant for most pesticides, is not included. For the risk assessment it should be relevant for this kind of study to conduct both acute and chronic assessments, using the Acute Reference Dose in addition to the ADI. The authors are encouraged to measure actual levels by conducting chemical analysis of representative samples and update the estimations.

Reviewer #2: The article entitled “Food: source of life or disease? Risk assessment of pesticide residue ingestion in food offered by institutional restaurants” is really an interesting topic to discuss but I found some major flaws in the article. Here are some key points to improve your article

Author cannot use question mark in the title.

99% ….there should be space between number and unit

In title write pesticide residues

In keyword write first letter capital ….Diet

Provide the full form of all the abbreviations

What is AI

There should be space between unit and word

If AI is active ingredient then replace it with a.i

What is IR

How you have detected that the pesticides are present in the food. Any data or analysis you have done.

How can you confidentially say that the food which is offered during the lunch is contaminated with pesticides, no doubt it unprocessed but until you have not done the proper analysis, you cannot say the food is contaminated with pesticides?

Which technique and procedure you have used for analysis.

Reviewer #3: The topic is of great relevance to public health and food safety, and the research contributes to a better understanding of the possible impacts of pesticides on human health. The work proposes research in an area with few studies developed in Brazil, which makes it difficult to discuss the data. However, the author used dubious bibliographies to support his point of view on the use of pesticides, such as references 7, 15, 56, and 85. Additionally, it is not possible to identify whether references 21 and 47 are the same and where they are available for access.

In lines 37 and 38, the author states that Brazil is "one of the leading countries in the world in the consumption of pesticides," but does not mention the bibliography on which this statement is based. Focus on presenting concrete data and scientific evidence about the risks of pesticide residues in chronic consumption. Avoid expressions and arguments that could be interpreted as activism or militancy. The goal is to inform readers based on verifiable facts, allowing them to draw their own conclusions.

In the objective of the work, the author proposes to evaluate foods and the risks of exposure. However, during the discussion, he does not develop arguments about the risk of exposure, and in the conclusion, he only highlights the role of collective meal restaurants in promoting nutritional safety. Critically review existing literature, highlighting both identified risks and gaps in current knowledge. There are already published works on pesticide residues in collective meals, such as in school lunches, which were not cited in the work. These similar works can help in the discussion of data.

It is important to highlight the pesticide residue and not the food itself, as is the case in reference 41 regarding residues present in beans. It is compared with work in Cameroon, which may not generate equivalence due to different planting techniques and pesticides authorized for use and MRL.

The discussion should address potential health risks in a clear and balanced way, without exaggerating the dangers or minimizing the impacts.

Suggestions:

Describe the mechanisms by which pesticide residues can affect human health.

Provide an analysis of exposure levels considered safe and compare them with levels found in the foods studied.

Include a discussion of the long-term effects of chronic consumption of pesticide residues, based on case studies, public health reports, and peer-reviewed scientific research.

In line 64, the author mentions that there is no data on orange foods in ANVISA monographs. These are included in the citrus group, so review pesticide residues considering this category.

Conclusions should reflect the results of the study in a balanced and evidence-based manner, avoiding recommendations that may appear excessively alarmist or unscientific.

Suggestions:

Summarize the main results of the study, highlighting the implications for public health.

Make recommendations for future food safety research and policy based on the evidence presented.

Regarding the title, one suggestion is to remove the question "Food: source of life or disease?" and leave only "Risk Assessment of Ingestion of Pesticide Residues in Food Offered by Institutional Restaurants." Adopting an objective, data-driven approach will strengthen your article's credibility and broaden its impact on the scientific community and society.

6. PLOS authors have the option to publish the peer review history of their article (what does this mean?). If published, this will include your full peer review and any attached files.

Reviewer #1: No

Reviewer #2: No

Reviewer #3: No

---

## [Author Response · Author response to Decision Letter 0]

16 Aug 2024

RESPONSE TO REVIEWERS

Dear Dr. Ulaganathan Arisekar (Academic Editor of the PLOS ONE)

We have submitted our revised manuscript titled “Food: source of life or disease? Risk assessment of pesticide residue ingestion in food offered by institutional restaurants” (PONE-D-24-15176) to PLOS ONE and are grateful and honored by the editorial decision for revisions. The review process has been thorough and coherent, and we have prepared this letter to address all the reviewers' comments. We respect all the feedback and decisions, and our aim is to clarify any doubts and reinforce the relevance of our study to public health. Our responses are highlighted in italic and grey marks. In addition to this letter, we have highlighted the changes in the manuscript text in yellow. Moreover, we are providing our database as supplementary material.We have sent our database as supporting materials.

Response to Reviewer #2 

Reviewer #2: The article entitled “Food: source of life or disease? Risk assessment of pesticide residue ingestion in food offered by institutional restaurants” is really an interesting topic to discuss but I found some major flaws in the article. Here are some key points to improve your article.

Author cannot use question mark in the title.

We have revised the title to make it clearer and more straightforward, removing any ambiguous elements. The revised title is: "Risk assessment of pesticide residues ingestion in food offered by institutional restaurant menus”

99% ….there should be space between number and unit

This has been corrected.

In title write pesticide residues

This has been corrected.

In keyword write first letter capital ….Diet

This has been corrected.

Provide the full form of all the abbreviations

A thorough review of all abbreviations and names was conducted.

What is AI

"AI" stands for active ingredient and was abbreviated as "a.i."

There should be space between unit and word

Spaces were added between units and words.

If AI is active ingredient then replace it with a.i

This has been corrected.

What is IR

"IR" is the abbreviation for institutional restaurants. However, to improve clarity, we standardized the term to "restaurants" throughout the text.

How you have detected that the pesticides are present in the food. Any data or analysis you have done.

Thank you for the question. We did not directly analyze any pesticide residues in the food samples. Instead, we applied a methodology to estimate the theoretical maximum daily intake (TMDI) and assessed the risk of exposure based on body weight. This methodology, validated by the World Health Organization (WHO), offers a reliable approach to estimating potential exposure to pesticide residues. By using this approach, we provide an amplified perspective on the potential chronic ingestion of pesticides, without the need for direct residue analysis. The data collection of the offered foods was carried out by the authors in person at the restaurants based on the available menus.

How can you confidentially say that the food which is offered during the lunch is contaminated with pesticides, no doubt it unprocessed but until you have not done the proper analysis, you cannot say the food is contaminated with pesticides?

Thank you for the question. Yes, we acknowledge that the information was generalized. However, based on our knowledge of pesticide use throughout the food production chain and the permissive legislation regarding various active ingredients, we can infer that the food is exposed to these chemical compounds. Nevertheless, we have chosen to state only what we have estimated as residue levels exceeding the Acceptable Daily Intake (ADI). As described below, the estimate of pesticide presence in the foods was made based on secondary data from analyses conducted by the National Health Surveillance Agency (ANVISA), a body of the Ministry of Health of Brazil.

Which technique and procedure you have used for analysis.

The method for calculating the Maximum Tolerable Daily Intake (MTDI) proposed by the WHO in 1997 involves estimating the theoretical maximum daily intake of the pesticide, based on its presence in food. The approach follows these steps:

1- Identification of pesticide in food offered in the menus: for this, we used the Brazilian pesticide database from ANVISA, and Codex Alimentarius database. 

2- Estimation of pesticide concentration: we obtained the average concentration of the pesticide in each food, based on MRL and the per quantity offered in the menus.

3- Estimation of food intake in the lunch meal: calculate the average amount of each food offered in the lunch menu daily by population groups.

4- Calculation of daily pesticide intake: multiply the concentration of the pesticide in each food by the intake amount of the lunch. 

5- Summation of intakes: Sum the various pesticide intakes from all offered foods to obtain the theoretical maximum daily intake.

6- Comparison with the Acceptable Daily Intake (ADI): Compare the theoretical maximum daily intake with the ADI established for the pesticide for evaluated exposed risks.

Response to Reviewer #3

Reviewer #3: The topic is of great relevance to public health and food safety, and the research contributes to a better understanding of the possible impacts of pesticides on human health. The work proposes research in an area with few studies developed in Brazil, which makes it difficult to discuss the data. (1) However, the author used dubious bibliographies to support his point of view on the use of pesticides, such as references 7, 15, 56, and 85. Additionally, it is not possible to identify whether (2) references 21 and 47 are the same and where they are available for access.

(1) The cited references were obtained from official websites with international credibility. However, we have chosen to remove these references without compromising the article.

(2) References 21 and 47 are not the same; they refer to studies conducted by the Consumer Defense Institute in Brazil on pesticide residues in ultra-processed foods. The first publication was in 2021, focusing on plant-based ultra-processed foods, and the second was in 2022, focusing on animal-based foods.

In lines 37 and 38, the author states that (3) Brazil is "one of the leading countries in the world in the consumption of pesticides," but does not mention the bibliography on which this statement is based. Focus on presenting concrete data and scientific evidence about the risks of pesticide residues in chronic consumption. (4) Avoid expressions and arguments that could be interpreted as activism or militancy. The goal is to inform readers based on verifiable facts, allowing them to draw their own conclusions.

(3) We added two references that support the information.

(4) We removed the few words that advocate for militancy and activism; and we also revised the text to avoid alarmism.

In the objective of the work, the author proposes to evaluate foods and the risks of exposure. However, during the (5) discussion, he does not develop arguments about the risk of exposure, and in the (6) conclusion, he only highlights the role of collective meal restaurants in promoting nutritional safety. Critically review existing literature, highlighting both identified risks and gaps in current knowledge. There are already published works on pesticide residues in collective meals, such as in school lunches, which were not cited in the work. These similar works can help in the discussion of data.

(5) Throughout the discussion of the study, we addressed the risks of pesticide exposure and health impacts; however, references supporting the discussion were added, such as those on the mechanisms of action of pesticides and their long-term effects from chronic intoxication.

(6) We have added in the discussion the study 'Pesticide residues potentially present in the regular diet of school children' to further support the findings of our research. We enhanced the conclusion by broadening the focus beyond institutional food to also address public health, including suggestions for future research and policy recommendations. 

It is important to highlight the pesticide residue and not the food itself, as is the case in (7) reference 41 regarding residues present in beans. It is compared with work in Cameroon, which may not generate equivalence due to different planting techniques and pesticides authorized for use and MRL.

(7) We added this important information. 

The discussion should address potential health risks in a clear and balanced way, without exaggerating the dangers or minimizing the impacts.

Suggestions:

(8) Describe the mechanisms by which pesticide residues can affect human health. Provide an analysis of exposure levels considered safe and compare them with levels found in the foods studied.

We have added two references that validates the mechanisms by which pesticides impact the occurrence of chronic diseases in humans. 

(9) Include a discussion of the long-term effects of chronic consumption of pesticide residues, based on case studies, public health reports, and peer-reviewed scientific research.

We added a paragraph about the exposition of the long-term effects of pesticides. 

(10) In line 64, the author mentions that there is no data on orange foods in ANVISA monographs. These are included in the citrus group, so review pesticide residues considering this category.

This was a mistake; the orange was analyzed under the category of citrus fruits. We have described this information in the methodology.

(11) Conclusions should reflect the results of the study in a balanced and evidence-based manner, avoiding recommendations that may appear excessively alarmist or unscientific.

Suggestions: Summarize the main results of the study, highlighting the implications for public health. Make recommendations for future food safety research and policy based on the evidence presented.

The conclusion text became more objective, focusing on the scientific evidence found and avoiding alarmism.

(12) Regarding the title, one suggestion is to remove the question "Food: source of life or disease?" and leave only "Risk Assessment of Ingestion of Pesticide Residues in Food Offered by Institutional Restaurants." Adopting an objective, data-driven approach will strengthen your article's credibility and broaden its impact on the scientific community and society.

Thank you for your suggestions. We have revised the title to "Risk assessment of pesticide residues ingestion in food offered by institutional restaurant menus”

---

## [Decision Letter · Decision Letter 1]

1 Nov 2024

Risk assessment of pesticide residues ingestion in food offered by institutional restaurant menus

PONE-D-24-15176R1

Dear Dr. ROLIM,

We’re pleased to inform you that your manuscript has been judged scientifically suitable for publication and will be formally accepted for publication once it meets all outstanding technical requirements.

Kind regards,

Ulaganathan Arisekar

Academic Editor

PLOS ONE

Additional Editor Comments (optional):

None

Reviewers' comments:

Reviewer's Responses to Questions

**Comments to the Author**

1. If the authors have adequately addressed your comments raised in a previous round of review and you feel that this manuscript is now acceptable for publication, you may indicate that here to bypass the “Comments to the Author” section, enter your conflict of interest statement in the “Confidential to Editor” section, and submit your "Accept" recommendation.

Reviewer #3: All comments have been addressed

2. Is the manuscript technically sound, and do the data support the conclusions?

Reviewer #3: Yes

3. Has the statistical analysis been performed appropriately and rigorously? 

Reviewer #3: Yes

4. Have the authors made all data underlying the findings in their manuscript fully available?

Reviewer #3: Yes

5. Is the manuscript presented in an intelligible fashion and written in standard English?

Reviewer #3: Yes

6. Review Comments to the Author

Reviewer #3: Congratulations to the authors for the revisions and for making the suggested changes. The subject is very relevant to the scientific community for discussion and reflection, and especially for the Brazilian population

7. PLOS authors have the option to publish the peer review history of their article (what does this mean?). If published, this will include your full peer review and any attached files.

Reviewer #3: No

---

## [Editor Report · Acceptance letter]

19 Nov 2024

PONE-D-24-15176R1 

PLOS ONE

Dear Dr. Rolim, 

I'm pleased to inform you that your manuscript has been deemed suitable for publication in PLOS ONE. Congratulations! Your manuscript is now being handed over to our production team.

Kind regards, 

on behalf of

Dr. Ulaganathan Arisekar 

Academic Editor

PLOS ONE